**Can mud (silt and clay) concentration be used to predict soil organic carbon**
**content within seagrass ecosystems?**
Oscar Serrano[1,2*], Paul S. Lavery[1,3], Carlos M. Duarte[4], Gary A. Kendrick[2,5], Antoni Calafat[6],
Paul York[7], Andy Steven[8], Peter Macreadie[9,10]
[1] School of Natural Sciences & Centre for Marine Ecosystems Research, Faculty of Health,
Engineering and Science, Edith Cowan University, Joondalup, Western Australia 6027
[2] The UWA Oceans Institute, The University of Western Australia, Crawley, WA, Australia
[3] Centro de Estudios Avanzados de Blanes, Consejo Superior de Investigaciones Científicas.
Blanes, Spain 17300
[4] Red Sea Research Center, King Abdullah University of Science and Technology, 4700
KAUST, Thuwal 23955-6900, Saudi Arabia,
[5] The School of Plant Biology, The University of Western Australia, Crawley, WA, Australia
[6] GRC Geociències Marines, Departament de Dinàmica de la Terra i de l'Oceà, Universitat de
Barcelona, Barcelona, Spain
[7] Centre for Tropical Water and Aquatic Ecosystem Research (TropWATER), James Cook
University, Cairns QLD 4870, Australia
[8] CSIRO, EcoSciences Precinct - Dutton Park 41 Boggo Road Dutton Park QLD 4102, Australia.
[9] Centre for Integrative Ecology, School of Life and Environmental Sciences, Deakin University,
Burwood, Victoria 3125, Australia
[10] Plant Functional Biology and Climate Change Cluster, University of Technology Sydney,
Broadway, New South Wales 2007, Australia
**\*Corresponding author:** Oscar Serrano (o.serranogras@ecu.edu.au)
**ABSTRACT**
The emerging field of blue carbon science is seeking cost-effective ways to estimate the organic
carbon content of soils that are bound by coastal vegetated ecosystems. Organic carbon ($C_{org}$)
content in terrestrial soils and marine sediments has been correlated with mud content (i.e. silt
and clay, particle sizes <63 µm), however, empirical tests of this theory are lacking for coastal
vegetated ecosystems. Here, we compiled data (n = 1345) on the relationship between $C_{org}$ and
mud contents in seagrass ecosystems (79 cores) and adjacent bare sediments (21 cores) to
address whether mud can be used to predict soil $C_{org}$ content. We also combined these data with
the $\delta^{13}C$ signatures of the soil $C_{org}$ to understand the sources of $C_{org}$ stores. The results showed
that mud is positively correlated with soil $C_{org}$ content only when the contribution of seagrass-
derived $C_{org}$ to the sedimentary $C_{org}$ pool is relatively low, such as in small and fast-growing
meadows of the genera *Zostera*, *Halodule* and *Halophila*, and in bare sediments adjacent to
seagrass ecosystems. In large and long-living seagrass meadows of the genera *Posidonia* and
*Amphibolis* there was a lack of, or poor relationship between mud and soil $C_{org}$ content, related to
a higher contribution of seagrass-derived $C_{org}$ to the sedimentary $C_{org}$ pool in these meadows.
The relative high soil $C_{org}$ contents with relatively low mud contents (e.g. mud-$C_{org}$ saturation) in
bare sediments and *Zostera*, *Halodule* and *Halophila* meadows was related to significant
allochthonous inputs of terrestrial organic matter, while higher contribution of seagrass detritus
in *Amphibolis* and *Posidonia* meadows disrupted the correlation expected between soil $C_{org}$ and
mud contents. This study shows that mud is not a universal proxy for blue carbon content in
seagrass ecosystems, and therefore should not be applied generally across all seagrass habitats.
Mud content can only be used as a proxy to estimate soil $C_{org}$ content for scaling up purposes
when opportunistic and/or low biomass seagrass species (i.e. *Zostera*, *Halodule* and *Halophila*)
are present (explaining 34 to 91% of variability), and in bare sediments (explaining 78% of the
variability). The results obtained could enable robust scaling up exercises at a low cost as part of
blue carbon stock assessments.

## 1. INTRODUCTION

The sedimentary organic carbon ($C_{org}$) stores of seagrass meadows – often referred to as 'blue carbon' – can vary among seagrass species and habitats, with reports of up to 18-fold differences (Lavery et al. 2013). Ambiguity remains in the relative importance of the depositional environment and species characteristics contributing to this variability. Seagrasses occur in a variety of coastal habitats, ranging from highly depositional environments to highly exposed and erosional habitats (Carruthers et al. 2007). Since seagrass species differ in their biomass and canopy structure, and occur in a variety of habitat types, this raises the question of whether mud content can be used to predict $C_{org}$ content within coastal sediments, or whether the species composition will significantly influence the soil $C_{org}$ stores independently of the geomorphological nature of the habitat.

Geomorphological settings (i.e. topography and hydrology), soil characteristics (e.g. mineralogy and texture) and biological features (e.g. primary production and remineralization rates) control soil $C_{org}$ storage in terrestrial ecosystems (Amundson, 2001, De Deyn et al. 2008; Jonsson and Wardle, 2009) and in mangrove and tidal salt marshes (Donato et al. 2011; Adame et al. 2013; Ouyang and Lee, 2014). While it is clear that habitat interactions have a large influence on stores of soil $C_{org}$, our understanding of the factors regulating this influence in seagrass meadows is limited (Nellemann et al. 2009; Duarte et al. 2010; Serrano et al. 2014).

The accumulation of $C_{org}$ in seagrass meadows results from several processes: accretion (autochthonous plant and epiphyte production, and trapping of allochthonous $C_{org}$; Kennedy et al. 2010), erosion (e.g. export; Romero and Pergent, 1992; Hyndes et al. 2014) and decomposition (Mateo et al. 1997). Previous studies demonstrate that both autochthonous (e.g. plant detritus and epiphytes) and allochthonous (e.g. macroalgae, seston and terrestrial matter) sources contribute

to the $C_{org}$ pool in seagrass soils (Kennedy et al. 2010; Watanabe and Kuwae, 2015). Plant net
primary productivity is a key factor controlling the amount of $C_{org}$ potentially available for
sequestration in seagrass ecosystems (Serrano et al. 2014), but the depositional environment is an
important factor controlling $C_{org}$ storage in coastal habitats (De Falco et al. 2004; Lavery et al.

80  2013).

Previous studies have shown a large variation in $C_{org}$ stores among morphologically different
seagrass species (Lavery et al. 2013; Rozaimi et al. 2013). Also, that $C_{org}$ accumulates more in
estuaries compared to coastal ocean environments (estimated at 81 Tg $C_{org}$ $y^{-1}$ and 45 Tg $C_{org}$ $y^{-1}$,
respectively; Nellemann et al. 2009). This is due largely to estuaries being highly depositional
environments, receiving fine-grained particles from terrestrial and coastal ecosystems which
enhance $C_{org}$ accumulation (i.e. silt and clay sediments retain more $C_{org}$ compared to sands; Keil
and Hedges, 1993; Burdige 2007) and preservation (i.e. reducing redox potentials and
remineralization rates; Hedges and Keil, 1995; Dauwe et al. 2001; Burdige, 2007; Pedersen et al.
2011). The inputs of seagrass-derived $C_{org}$ in the sedimentary pool could break the linear
relationship among mud (i.e. silt and clay particles) and $C_{org}$ contents typically found in
terrestrial (Nichols, 1984; McGrath and Zhang, 2003) and marine sedimentary environments
(Bergamaschi et al. 1997; De Falco et al. 2004). However, the amount of $C_{org}$ that can be
associated with mud particles is limited (Hassink, 1997), which could lead to a poor relationship
between mud and soil $C_{org}$ contents. Also, other factors found to play a key role in controlling
soil $C_{org}$ accumulation in terrestrial and coastal ecosystems, such as chemical stabilization of
organic matter (Percival et al. 1999; Burdige, 2007), carbon in microbial biomass (Sparling,
1992; Danovaro et al. 1995), and soil temperature (Pedersen et al. 2011), could also influence
$C_{org}$ storage in seagrass meadows.
A significant relationship between mud and $C_{org}$ contents would allow mud to be used as a
proxy for $C_{org}$ content, thereby enabling robust scaling up exercises at a low cost as part of blue
carbon stock assessments. Furthermore, since most countries have conducted geological surveys
within the coastal zone to determine sediment grain size, a strong, positive relationship between
mud and $C_{org}$ contents would allow the development of geomorphology models to predict blue
carbon content within seagrass meadows, dramatically improving global estimates of blue carbon
storage. The purpose of this study was therefore to test for relationships between $C_{org}$ and mud
contents within seagrass ecosystems and adjacent bare sediments.

## 108   2. MATERIAL AND METHODS

Data was compiled from a number of published and unpublished studies from Australia and
Spain, in seagrass meadows across diverse habitats (Table 1). The study sites encompass
monospecific and/or mixed meadows from a variety of temperate and tropical seagrass species of
the genera *Posidonia*, *Amphibolis*, *Zostera*, *Halophila* and *Halodule*, and adjacent bare
sediments, while including a variety of depositional environments (from estuarine to exposed
coastal areas encompassing different water depths, from intertidal to the deep limit of seagrass
distribution; Table 1). Data from 100 cores (79 from seagrass meadows and 21 from bare
sediments) on sediment grain size, organic carbon ($C_{org}$) content and stable carbon isotope
signatures of the $C_{org}$ ($\delta^{13}C$) was explored in this study (N = 1345).
Sediment cores were sampled by means of percusion and rotation, or vibrocoring (ranging
from 10 to 475 cm long). The core barrels consisted of PVC or aluminium pipes (50 to 90 mm
inside diamater) with sharpenned ends to cut fibrous material and minimize core shortening
(compression) during coring (Serrano et al. 2012, 2014). All cores were sealed at both ends,
transported vertically to the laboratory and stored at 5°C before processing.
The cores were sliced at regular intervals, each slice/sample was weighed before and after
oven drying to constant weight at 70°C (DW), and subsequently sub-divided for analysis. The
$C_{org}$ elemental and isotopic composition of the organic matter was measured in milled
subsamples from several slices along the cores. The sediment core sub-samples were acidified
with 1 M HCl, centrifuged (3500 RPM; 5 minutes) and the supernatant with acid residues was
removed using a pipette, then washed in deionized water, centrifuged again and the supernatant
removed. The residual samples were re-dried (70°C) before carbon elemental and isotopic
analyses. The samples were encapsulated and the organic carbon elemental and isotopic
composition was analyzed using an elemental analyzer interfaced with an isotope ratio mass
spectrometer. Percentage $C_{org}$ was calculated for the bulk (pre-acidified) samples. Carbon isotope
ratios are expressed as δ values in parts per thousand (‰) relative to VPDB (Vienna Pee Dee
Belemnite). For sediment grain size analysis, a Coulter LS230 laser-diffraction particle analyzer
was used following digestion of the samples with 10% hydrogen peroxide. The mud content in
the sediments (<63 μm) was determined, and expressed as a percentage of the bulk sample.
Pearson correlation analysis was used to test for significant relationships among $C_{org}$ and
mud contents, and $C_{org}$ and $\delta^{13}C$ signatures. Correlations between the variables studied were
tested among seagrass species (9 categories) and bare sediments, seagrass genera (4 categories),
habitat geomorphology (coastal and estuarine habitats) and soil depth (in 1 to 10 cm-thick and 11
to 110 cm-thick deposits).

**3. RESULTS**
The soil organic carbon ($C_{org}$) and mud contents varied within the seagrass meadows and
bare sediments studied in Australia and Spain. The soil $C_{org}$ and mud contents were higher in
seagrass meadows (average ± SE, 1.5 ± 0.2% and 18 ± 2.4%, respectively) compared to bare
sediments (0.6 ± 0.1% and 10.8 ±1 .2%, respectively; Table 2). On average, seagrass meadows
of the genera *Amphibolis* and *Posidonia* contained higher soil $C_{org}$ (1.6 ± 0.1%) and lower mud
(7.2 ± 0.4) than meadows of *Halophila*, *Halodule* and *Zostera* (1.2 ± 0.2% and 34.9 ± 5.4%,
respectively; Table 2). Overall, carbon isotopic ratios from sedimentary organic matter ($\delta^{13}$C)
were similar between seagrass soils and bare sediments (-17.6 ± 0.3‰ and -17.3 ± 0.2‰,
respectively). The $C_{org}$ in soils from *Posidonia* and *Amphibolis* meadows were $^{13}$C-enriched (-
15.5 ± 0.3‰) compared with seagrass soils from *Halophila*, *Halodule* and *Zostera* meadows (-
20.7 ± 0.4‰; Table 2). The $C_{org}$ content in soils from estuarine and coastal habitats were similar,
while mud content in estuarine sediments was higher and $\delta^{13}$C values depleted when compared
to coastal habitats (Table 2).
The relationships between the variables studied (i.e. %$C_{org}$, %mud, and $\delta^{13}$C signatures of
sedimentary $C_{org}$) among different species and habitat geomorphologies, and among different soil
depths were explored in Figures 1 to 3, and Table 3. When accounting for the whole dataset (up
to 475 cm long cores), the $C_{org}$ content increased with increasing mud content in bare sediments
($R^2$ = 0.78) and at species level, except for *Posidonia oceanica* (i.e. $C_{org}$ content decreased with
increasing mud content; $R^2$ = 0.15) and *Amphibolis griffithii* (i.e. no relationship was found, $R^2$ =
0.05; Table 3). Although most of the correlations at species level were significant, they only
explain 2 to 39% of the variance in trends described, except for *Halophila ovalis* (91%; Table 3).
In particular, *Posidonia* meadows (*P. australis*, *P. sinuosa* and *P. oceanica*) had the lower
correlation values ($R^2$ ranged from 0.02 to 0.15). When combining mud and $C_{org}$ contents in
seagrass meadows of the colonizing and opportunistic genera *Halophila*, *Halodule* and *Zostera*
(Kilminster et al. 2015), a relatively high correlation was found ($R^2 = 0.56$; Figure 1), while soil
$C_{org}$ and mud contents in persistent genera were only slightly positively correlated in combined
*Amphibolis* spp and not correlated in *Posidonia* spp meadows (Figure 1).
The relationships between soil $C_{org}$ and mud contents within different core depths (from 1 to
10 cm-thick deposits, and from 11 to up to 110 cm-thick deposits) for bare sediments and each
group of seagrass species were explored in Figure 2. The $C_{org}$ content increased with increasing
mud content in bare sediments for both 1 to 10 cm-thick ($R^2 = 0.74$) and 11 to 110 cm-thick ($R^2$
$= 0.81$) soils. When combining mud and $C_{org}$ contents in seagrass meadows of the genera
*Halophila*, *Halodule* and *Zostera*, a higher correlation was found for deeper core sections (11 to
110 cm-thick; $R^2 = 0.74$) compared to top core sections (1 to 10 cm-thick; $R^2 = 0.17$). For
combined *Amphibolis* and *Posidonia* species, soil $C_{org}$ and mud contents were only slightly
positively correlated in deeper *Amphibolis* spp sections (11 to 110 cm-thick; $R^2 = 0.23$) and not
correlated in *Posidonia* spp meadows (Figure 2). The classification of habitats based on
geomorphology (i.e. coastal and estuarine) showed a lack of correlation between soil $C_{org}$ and
mud contents in coastal ecosystems, and a poor correlation in estuarine ecosystems ($R^2 = 0.14$;
Figure 3 and Table 3).
The relationships between soil $\%C_{org}$ and $\delta^{13}C$ signatures were poor for all individual
*Amphibolis* and *Posidonia* species studied ($R^2$ ranging from 0.09 to 0.3; Table 3), and for
combined *Amphibolis* spp (Figure 1), with a tendency of $C_{org}$-rich soils being enriched in $^{13}C$
(Figure 1). In contrast, $\%C_{org}$ and $\delta^{13}C$ signatures were not correlated in any of the small and
fast-growing *Halodule*, *Zostera*, *Halophila* meadows studied (Table 3), neither individually nor
when combined (Figure 1 and Table 3). A lack of correlation between soil $\%C_{org}$ and $\delta^{13}C$
signatures was also found in bare sediments adjacent to seagrass meadows (Figure 3 and Table

191    3).


## 4. DISCUSSION

Overall mud content is a poor predictor of soil $C_{org}$ in seagrass meadows and care should be
taken in its use as a cost-effective proxy or indicator of $C_{org}$ for scaling-up purposes in the
emerging field of blue carbon science. Although we describe some promise for opportunistic and
early colonizing *Halophila*, *Halodule* and *Zostera* meadows (i.e. mud content explained 34 to
91% of variability in $C_{org}$ content) and in bare sediments adjacent to seagrass meadows
(explaining 78% of the variability), mud is not a universal proxy for blue carbon content and
therefore should not be applied generally across all seagrass habitats. In particular, mud content
only explained 5 to 32% of soil $C_{org}$ content in *Amphibolis* spp meadows and 2 to 15% of soil
$C_{org}$ content in *Posidonia* spp meadows, and therefore, mud content is not a good proxy for blue
carbon content in these meadows.
A tenet of carbon cycling within the coastal ocean is that fine-grained sediments (i.e. mud)
have higher $C_{org}$ contents. The positive relationship found between mud and $C_{org}$ contents in
coastal bare sediments (explaining 78% of the variability) is in agreement with previous studies
(e.g. Bergamaschi et al. 1997; De Falco et al. 2004), and is related to their larger surface areas
compared to coarse-grained sediments, providing more binding sites for $C_{org}$ on the surface of
minerals (Keil and Hedges, 1993; Mayer, 1994a, 1994b; Galy et al. 2007; Burdige 2007). In
addition, the predominance of fine sediments reduces oxygen exchange and results in low
sediment redox potentials and remineralization rates, contributing to the preservation of
sedimentary $C_{org}$ after burial (Hedges and Keil, 1995; Bergamaschi et al. 1997; Dauwe et al.
2001; Burdige 2007; Pedersen et al. 2011). However, the maximum capacity of a given soil to
preserve $C_{org}$ by their association with clay and silt particles is limited (i.e. mud-$C_{org}$ saturation;
Hassink, 1997). The results obtained showed that bare sediment samples with relative high $C_{org}$
contents (i.e. >4% $C_{org}$) and relatively low mud contents were also $^{13}C$-depleted (Figure 1),
suggesting significant contributions of soil $C_{org}$ from allochthonous sources (e.g. terrestrial and
sestonic; Kennedy et al. 2010). This could have disrupted the correlation found between soil $C_{org}$
and mud contents in the bare sediments studied.

220         Mud is not a universal proxy for soil $C_{org}$ content in seagrass meadows, which could be

mainly explained by additional inputs of seagrass-derived $C_{org}$ and/or allocthonous $C_{org}$ to the
sedimentary $C_{org}$ pool, obviating the linear relationship between mud and $C_{org}$ contents found in
the absence of vegetation. The $\delta^{13}C$ values indicated that both seagrass-$C_{org}$ and non-seagrass-
derived $C_{org}$ (i.e. epiphytes, algae, seston or terrestrial matter) were buried in the soils of all
studied meadows, but are consistent with a model of increasing capture of seagrass-derived $C_{org}$
at meadows formed by persistent, high-biomass seagrasses (i.e. genera *Posidonia* and
*Amphibolis*) relative to opportunistic, low-biomass seagrasses (i.e. genera *Halophila*, *Halodule*
and *Zostera*).

229         On one hand, the soil $\delta^{13}C$ signatures measured in these long-living and large seagrass

meadows (averaging -15 ± 0.2‰ in both cases) were closer to the $\delta^{13}C$ signatures of *Posidonia*
and *Amphibolis* tissues (ranging from -8 to -14‰; Hyndes and Lavery 2005; Hindell et al. 2004;
Cardona et al. 2007; Fourqurean et al. 2007; Collier et al. 2008; Kennedy et al. 2010; Hanson et
al. 2010; Serrano et al. 2015) than to $\delta^{13}C$ values of algae or terrestrial organic matter (ranging
from -18 to -32‰; e.g. Smit et al. 2006; Cardona et al. 2007; Kennedy et al. 2010; Hanson et al
2010; Deudero et al. 2011). The poor relationship between mud and soil $C_{org}$ contents in
*Amphibolis* soils could be explained by samples with relative high $C_{org}$ contents (i.e. >2.5% $C_{org}$)
and relatively low mud contents, as a result of both the contribution of seagrass-derived $C_{org}$ (i.e.
$^{13}$C-enriched) and $C_{org}$ from allochthonous sources (i.e. $^{13}$C-depleted; Figure 1). In *Posidonia*
soils, the poor relationship between mud and soil $C_{org}$ contents could be explained by samples
with relative high $C_{org}$ contents (i.e. >10% $C_{org}$) and relatively low mud contents, as a result of
the contribution of seagrass-derived $C_{org}$ (i.e. $^{13}$C-enriched; Figure 1). The contribution of
seagrass-derived $C_{org}$ (i.e. root, rhizome and sheath detritus) in *Posidonia* soils play a much
larger role than the accumulation of fine, organic-rich allochthonous particles.
On the other hand, the soil $\delta^{13}$C signatures measured in *Halodule*, *Halophila* and *Zostera*
meadows (averaging -21 ± 0.4‰) were more similar to $\delta^{13}$C values of algae or terrestrial organic
matter than to $\delta^{13}$C values of their seagrass tissues (ranging from -10 and -14‰; e.g. Hemminga
and Mateo, 1996; Kennedy et al. 2010; Hanson et al. 2010). The positive relationship between
mud and soil $C_{org}$ contents in *Halodule*, *Halophila* and *Zostera* soils could be explained their
relatively high mud content and $^{13}$C-depleted $C_{org}$, indicating that allochthonous $C_{org}$ inputs and
mud content play a major role in soil $C_{org}$ accumulation in these opportunistic and early-
colonizing seagrasses. However, the relative high $C_{org}$ contents found with relatively low mud
contents (i.e. mud-$C_{org}$ saturation) disrupted the correlation found between soil $C_{org}$ and mud
contents in these meadows ($C_{org}$ >1% in samples with 0-20% mud; $C_{org}$ >2% in samples with 20-
70% mud and $C_{org}$ >3.5 in samples with 70-100% mud; Figure 1).
The results obtained showed a tendency for high-biomass and persistent meadows (i.e.
*Posidonia* and *Amphibolis*) to accumulate higher $C_{org}$ stores and seagrass-derived $C_{org}$ compared
to ephemeral and low-biomass meadows (i.e. *Halophila*, *Halodule* and *Zostera*), suggesting that
factors (biotic and abiotic) affecting the production, form and preservation of $C_{org}$ within habitats
exert a significant influence on soil $C_{org}$ content (Lavery et al. 2013; Serrano et al. 2014, 2015).
The above- and belowground biomass in meadows of the genus *Posidonia* (averaging 535 and
910 g DW m$^{-2}$, respectively) is up to 2-fold higher than in *Amphibolis* meadows (averaging 641
and 457 g DW m$^{-2}$, respectively) and 4 to 18-fold higher than in small and opportunistic
seagrasses of the genera *Halophila*, *Halodule* and *Zostera* (125 and 49 g DW m$^{-2}$, on average;
respectively; Duarte and Chiscano, 1999; Paling and McComb 2000). Indeed, larger seagrasses
tend to have larger and more persistent rhizomes, constituted by more refractory forms of $C_{org}$,
more prone to be preserved in soils than simpler, more labile forms of $C_{org}$ such as seston and
algal detritus which are more suitable to experience remineralization during early diagenesis
(Henrichs 1992; Burdige, 2007). In addition, the larger size of detritus within *Amphibolis* and
*Posidonia* meadows compared to *Halophila*, *Halodule* and *Zostera* meadows could also
contribute to the larger accumulation of $C_{org}$ in the former, since decay rates of seagrass detritus
increase with decreasing particle size due to larger surfaces available for microbial attack
(Harrison, 1989). Differences in above- and belowground biomass and recalcitrance between
*Posidonia* and *Amphibolis* spp could explain the larger contribution of seagrass-derived $C_{org}$ (i.e.
$^{13}$C-enriched) in the former, thereby obviating the linear relationship between mud and $C_{org}$
contents (Figure 1).
The soil $C_{org}$ content tend to decrease with soil depth and ageing in seagrass ecosystems (e.g.
Serrano et al. 2012), thereby the persistence of discrete organic detritus within upper soil
horizons could lead to organic matter concentrations above those levels explained by the
association with clay and silt particles, as previously demonstrated for terrestrial soils (Mayer
and Xing, 2001; Gami et al. 2009). The organic matter preserved in most marine sediments is
intimately associated with mineral surfaces (i.e. selective preservation by sorption of organic
matter into minerals; Keil et al 1994) and therefore the correlation between soil $C_{org}$ and mud
contents in seagrass meadows could vary as a function of soil depth and ageing. The results
obtained show that soil depth is not an important factor when attempting to predict soil $C_{org}$
content based on mud content in bare sediments (i.e. $R^2 > 0.74$ for all core depths explored; 1 to
110 cm-thick, 1 to 10 cm-thick, and 11 to 110 cm-thick; Figure 2). However, a clearer pattern
appeared when exploring the correlation between soil $C_{org}$ and mud contents in top 10 cm and
within 11-110 cm soil depths of combined *Halodule*, *Halophila* and *Zostera* species ($R^2 = 0.17$
and $R^2 = 0.74$, respectively). These results suggest that the relatively small belowground biomass
of these species (i.e. organic detritus) only has an impact on the expected positive correlation
between soil $C_{org}$ and mud content within the top 10 cm, while the correlation for deeper soil
depths (11-110 cm) improved ($R^2 = 0.74$) compared to the whole dataset (1 to 110 cm-thick; $R^2$
$= 0.56$). For combined *Amphibolis* and *Posidonia* species, the results obtained show that soil
depth is not an important factor when attempting to predict soil $C_{org}$ content based on mud
content (i.e. $R^2 < 0.2$ in all cases; 1 to 110 cm-thick, 1 to 10 cm-thick, and 11 to 110 cm thick;
Figure 2). These results suggest that the relatively large belowground biomass of these species
(i.e. organic detritus) has an impact on the expected positive correlation between soil $C_{org}$ and
mud content within all depths studied.
Habitat conditions in seagrass meadows not only influence the amount of $C_{org}$ accumulation
through detrital plant inputs, but the capacity of the plant canopies to retain particles (Gacia et al.
1999). The amount of fine suspended particles available for burial varies among sites, driven by
geomorphological features (e.g. run-off, hydrodynamic energy and water depth), while meadow
structure (i.e. density, cover and morphology of the canopy) constrains their capacity to
accumulate sediment particles (Hendriks et al. 2010; Peralta et al. 2008). Although the number of
cores and species studied in coastal and estuarine ecosystems was unbalanced (i.e. *Amphibolis*
and *Posidonia* dominate in coastal habitats and *Halophila*, *Halodule*, *Zostera* dominate in
estuarine habitats), the lack of, or poor correlations found within estuarine and coastal
ecosystems, precludes the general use of mud as a predictor of blue carbon content based on
habitat geomorphology (Figure 3). Seagrass meadows and bare sediments in environments
conducive for depositional processes (i.e. estuaries) accumulated up to 4-fold higher amounts of
mud compared to other coastal ecosystems, but the saturation of mud with $C_{org}$ and the large
contribution of seagrass detritus into the sedimentary $C_{org}$ pool ($^{13}C$-enriched soils) in some study
sites disrupted the positive relationship expected between mud and soil-$C_{org}$ contents. In
estuarine ecosystems, soil $C_{org}$ originated from both mud inputs linked to allochthonous-$C_{org}$ via
deposition from upstream transport (e.g. Aller, 1998) and seagrass inputs (i.e. in samples with
$C_{org}$ >5%; Figure 3). The insignificant relationship between mud and soil $C_{org}$ contents in coastal
habitats could be explained by their relatively low mud content and the accumulation of
seagrass-derived $C_{org}$, in particular in samples with $C_{org}$ >5% (Figure 3).
In sum, mud is not a universal proxy for blue carbon content in seagrass ecosystems and
should not be applied generally across all habitat and vegetation types. Overall, the positive
relationship between mud and $C_{org}$ contents found in bare sediments and in opportunistic and/or
low biomass seagrass meadows (i.e. genera *Zostera*, *Halodule* and *Halophila*) allow mud to be
used as a proxy for $C_{org}$ content in these ecosystems, thereby enabling robust scaling up exercises
(i.e. benefiting from existing geological surveys and models) at low cost as part of blue carbon
stock assessment programs. However, mud content is not a good predictor of $C_{org}$ content in
highly productive meadows such as those constituted by *P. oceanica* in the Mediterranean Sea
and *P. australis, P. sinuosa* and *Amphibolis* spp in Australia. Analyses of soil grain size (i.e.
%mud) could constitute a relatively cheap method to estimate soil organic carbon content in
seagrass ecosystems, particularly dry and wet sieving using standard geological sieves
(Erftemeijer and Kach, 2001). These could be used to cheaply quantify mud content as a proxy
for carbon, particularly in student projects, citizen science and in countries where funding for
science is limited and they do not have access to higher technology methods or cannot afford to
pay for analysis. In addition, since most countries have conducted geological surveys within the
coastal zone to determine sediment grain size (e.g. Passlow et al. 2005), a strong, positive
relationship between mud and $C_{org}$ contents could allow the development of geomorphology
models to predict blue carbon content within seagrass meadows, dramatically improving global
estimates of blue carbon storage. Indeed, maps of soil grain-size could be obtained using remote
sensing (Rainey et al. 2003; De Falco et al. 2010), opening new opportunities for scaling
exercises.
Previous studies suggested that the relationship between organic matter and the sediment
matrix is best seen with clay-sized fractions (<0.004 mm; Bergamaschi et al., 1997; De Falco et
al. 2004). However, the grain size cut-off selected in this study (mud, <0.063 mm) is more
representative of the bulk soil and their $C_{org}$ content (Pedrosa-Pàmies et al. 2013) and therefore a
higher correlation is expected when comparing bulk soil $C_{org}$ with a larger and more
representative fraction of the sediment (i.e. including the silt fraction, 0.004-0.063 mm, also
provides binding sites for $C_{org}$; Burdige, 2007). Other biological, chemical and geological factors
not explored in detail in this study may also play a key role in $C_{org}$ storage, and ultimately in the
relationship between soil $C_{org}$ and mud contents. For example, the effects of habitat
geomorphology (e.g. hydrodynamic energy, terrestrial mud and $C_{org}$ inputs, export of seagrass
biomass) and species identity (e.g. variation in terms of productivity, oxygen exposure and
recalcitrance of $C_{org}$ stores, and plant influence on sediment retention) within both coastal and
estuarine environments, are among the factors identified in this study which might explain
significant variation in the $C_{org}$ stores of meadows in relatively similar exposure conditions
(Serrano et al. 2015). Other factors found to play a key role in controlling soil $C_{org}$ accumulation
in terrestrial ecosystems, such as chemical stabilization of organic matter (Percival et al. 1999;
Galy et al. 2008) and microbial biomass carbon (Danovaro et al. 1994), could also influence $C_{org}$
storage in seagrass ecosystems. Further studies are needed to identify the influences of these
other factors on $C_{org}$ storage in seagrass meadows, and in addition to the mud content, other
characteristics should be taken into account when attempting to obtain robust estimates of $C_{org}$
stores within coastal areas.

**ACKNOWLEDGMENTS**
The raw data compiled for this study was published in ACEF Coastal Data portal (*DOI to be*
*provided*). This work was supported by the ECU Faculty Research Grant Scheme, the ECU Early
Career Research Grant Scheme, and the CSIRO Flagship Marine & Coastal Carbon
Biogeochemical Cluster (Coastal Carbon Cluster) with funding from the CSIRO Flagship
Collaboration Fund. PM was supported by an ARC DECRA DE130101084. The authors are
grateful to M. Rozaimi, A. Gera, P. Bouvais, A. Ricart, C. Bryant, G. Skilbeck, M. Rozaimi, A.
Esteban, M. A. Mateo, P. Donaldson, C. Sharples and R. Mount for their help in field and/or
laboratory tasks.

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

**Tables and Figures**

**Table 1.** Data on soil organic carbon and mud contents, and stable carbon isotope from coastal
soils were gathered from a variety of seagrass meadows (and also from adjacent bare sediments)
and habitat types.

| Species | Study site | Geomorphology | Number of cores | Number of samples | Core depth (cm) | Water depth (m) |
|---------|-----------|---------------|-----------------|-------------------|-----------------|-----------------|
| *Amphibolis* (mixed spp) | Rottnest Island, WA, Australia | Coastal | 2 | 68 | 0-120 | 2 |
| | Shark Bay, WA, Australia | Coastal | 1 | 38 | 0-170 | 2 |
| *Amphibolis antarctica* | Shark Bay, WA, Australia | Coastal | 2 | 63 | 0-200 | 2-3 |
| *Amphibolis griffithii* | Jurien Bay, WA, Australia | Coastal | 2 | 41 | 0-70 | 4 |
| *Posidonia australis* | Oyster Harbour, WA, Australia | Estuarine | 3 | 31 | 0-120 | 2 |
| | Waychinicup Inlet, WA, Australia | Estuarine | 2 | 79 | 0-150 | 2 |
| | Robbins Island, TAS, Australia | Coastal | 6 | 138 | 0-180 | 3 |
| *Posidonia sinuosa* | Frenchman's Bay, WA, Australia | Coastal | 4 | 100 | 0-80 | 2-8 |
| | Cockburn Sound, WA, Australia | Coastal | 3 | 50 | 0-30 | 6 |
| | Garden Island, WA, Australia | Coastal | 5 | 147 | 0-120 | 2-8 |
| *Posidonia oceanica* | Portlligat, Spain | Coastal | 1 | 192 | 475 | 3 |
| | Balearic Islands, Spain | Coastal | 6 | 25 | 0-270 | 3 |
| *Halodule uninvervis* | Carnarvon, WA, Australia | Estuarine | 1 | 39 | 0-210 | 2 |
| | Gladstone, QLD, Australia | Estuarine | 6 | 6 | 0-10 | intertidal |
| *Halophila decipiens* | Gladstone, QLD, Australia | Estuarine | 2 | 2 | 0-10 | intertidal |
| *Halophila ovalis* | Rottnest Island, WA, Australia | Coastal | 1 | 17 | 0-30 | 3 |
| | Swan River, WA, Australia | Estuarine | 1 | 5 | 0-70 | 2 |
| | Leschenault Inlet, WA, Australia | Estuarine | 1 | 8 | 0-120 | 1 |
| | Harvey Inlet, WA, Australia | Estuarine | 1 | 5 | 0-20 | 2 |
| | Gladstone, QLD, Australia | Estuarine | 2 | 2 | 0-10 | intertidal |
| *Zostera muelleri* | Fagans Bay, NSW, Australia | Estuarine | 2 | 20 | 0-10 | intertidal |
| | Gladstone, QLD, Australia | Estuarine | 23 | 23 | 0-10 | intertidal |
| | Tuggerah Lakes, NSW, Australia | Estuarine | 2 | 64 | 0-400 | 3 |
| Bare | Cockburn Sound, WA, Australia | Coastal | 10 | 131 | 0-30 | 2-9 |
| | Garden Island, WA, Australia | Coastal | 1 | 16 | 0-30 | 4 |
| | Oyster Harbour, WA, Australia | Estuarine | 1 | 26 | 0-110 | 3 |
| | Gladstone, QLD, Australia | Estuarine | 9 | 9 | 0-10 | intertidal |






**Table 2.** Average ± SE organic carbon ($C_{org}$) content (in %), $\delta^{13}C$ signatures and mud content in
all habitats and soil depths studied. a) Descriptive statistics based on species identity. b)
Descriptive statistics based on habitat geomorphology (estuarine *vs* coastal environments). N,
number of samples.

a)

| Habitat (species) | Organic carbon (%) | | | $\delta^{13}C$ (‰) | | | Mud (%) | | |
|---|---|---|---|---|---|---|---|---|---|
| | N | Mean | SE | N | Mean | SE | N | Mean | SE |
| *Posidonia oceanica* | 217 | 3.91 | 0.35 | 217 | -14.92 | 0.08 | 217 | 11.73 | 0.53 |
| *Posidonia australis* | 248 | 1.87 | 0.08 | 244 | -15.79 | 0.24 | 248 | 11.79 | 0.68 |
| *Posidonia sinuosa* | 297 | 0.80 | 0.04 | 291 | -14.08 | 0.16 | 297 | 2.59 | 0.18 |
| *Amphibolis* (mixed spp) | 106 | 1.41 | 0.11 | 106 | -15.20 | 0.23 | 106 | 4.75 | 0.33 |
| *Amphibolis antarctica* | 63 | 0.99 | 0.06 | 62 | -14.62 | 0.24 | 63 | 6.64 | 0.44 |
| *Amphibolis griffithii* | 41 | 0.85 | 0.07 | 36 | -15.83 | 0.56 | 41 | 5.44 | 0.29 |
| *Halodule uninervis* | 45 | 0.78 | 0.12 | 45 | -19.86 | 0.53 | 45 | 17.68 | 3.04 |
| *Zostera muelleri* | 107 | 1.10 | 0.07 | 43 | -20.02 | 0.30 | 107 | 31.68 | 2.59 |
| *Halophila decipiens* | 2 | 1.87 | 0.51 | 2 | -25.60 | 0.31 | 2 | 65.99 | 9.62 |
| *Halophila ovalis* | 37 | 0.97 | 0.23 | 37 | -17.22 | 0.44 | 37 | 24.09 | 6.23 |
| Bare | 182 | 0.59 | 0.08 | 182 | -17.25 | 0.24 | 182 | 10.83 | 1.20 |
| **Grand Total** | 1345 | 1.56 | 0.07 | 1265 | -16.18 | 0.10 | 1345 | 10.83 | 0.43 |

b)

| Habitat (geomorphology) | Organic carbon (%) | | | $\delta^{13}C$ (‰) | | | Mud (%) | | |
|---|---|---|---|---|---|---|---|---|---|
| | N | Mean | SE | N | Mean | SE | N | Mean | SE |
| Coastal | 1026 | 1.59 | 0.09 | 1014 | -15.70 | 0.10 | 1026 | 6.85 | 0.24 |
| Estuarine | 319 | 1.44 | 0.07 | 251 | -18.10 | 0.24 | 319 | 23.62 | 1.41 |




**Table 3.** Pearson correlation analyses to test for significant relationships among soil $C_{org}$ and
mud contents, and soil $C_{org}$ and $\delta^{13}C$ signatures in up to 475 cm long cores; based on (a) species
identity and (b) habitat geomorphology. *ns*, non significant correlation.

a)

| Habitat (species) | Organic carbon (%) vs mud (%) | | | Organic carbon (%) vs $\delta^{13}C$ (‰) | | |
|---|---|---|---|---|---|---|
| | Formula | $R^2$ | P value | Formula | $R^2$ | P value |
| *Posidonia oceanica* | $C_{org} = -0.26*mud + 6.95$ | 0.15 | *** | $C_{org} = 1.59*\delta^{13}C + 27.61$ | 0.13 | *** |
| *Posidonia australis* | $C_{org} = 0.02*mud + 1.69$ | 0.02 | * | $C_{org} = 0.18*\delta^{13}C + 4.73$ | 0.30 | *** |
| *Posidonia sinuosa* | $C_{org} = 0.07*mud + 0.61$ | 0.09 | *** | $C_{org} = 0.12*\delta^{13}C + 2.44$ | 0.23 | *** |
| *Amphibolis* (mixed spp) | $C_{org} = 0.17*mud + 0.61$ | 0.26 | *** | $C_{org} = 0.14*\delta^{13}C + 3.53$ | 0.09 | ** |
| *Amphibolis antarctica* | $C_{org} = 0.08*mud + 0.47$ | 0.32 | *** | $C_{org} = 0.14*\delta^{13}C + 3.10$ | 0.29 | *** |
| *Amphibolis griffithii* | *ns* | 0.05 | 0.18 | $C_{org} = 0.06*\delta^{13}C + 1.79$ | 0.21 | ** |
| *Halodule uninervis* | $C_{org} = 0.02*mud + 0.37$ | 0.34 | *** | *ns* | 0.00 | 0.89 |
| *Zostera muelleri* | $C_{org} = 0.02*mud + 0.54$ | 0.39 | *** | *ns* | 0.08 | 0.07 |
| *Halophila ovalis* | $C_{org} = 0.04*mud + 0.12$ | 0.91 | *** | *ns* | 0.00 | 0.89 |
| Bare | $C_{org} = 0.06*mud - 0.03$ | 0.78 | *** | *ns* | 0.01 | 0.24 |

b)

| Habitat (geomorphology) | Organic carbon (%) vs mud (%) | | | Organic carbon (%) vs $\delta^{13}C$ (‰) | | |
|---|---|---|---|---|---|---|
| | Formula | $R^2$ | P value | Formula | $R^2$ | P value |
| Coastal | *ns* | 0.01 | 0.85 | $C_{org} = 0.17*\delta^{13}C + 4.14$ | 0.03 | *** |
| Estuarine | $C_{org} = 0.02*mud + 1.01$ | 0.14 | * | $C_{org} = 0.17*\delta^{13}C + 4.52$ | 0.22 | ** |





**Figure 1.** Relationships among soil $C_{org}$ and mud contents, and soil $C_{org}$ and $\delta^{13}C$ signatures in
all habitats and all soil depths studied: bare sediments, combined *Halodule*, *Halophila* and
*Zostera* species, and combined *Amphibolis* and *Posidonia* species. Only correlations with $R^2$
>0.5 are shown. The grey shaded areas showed the range of $\delta^{13}C$ signatures of plant detritus
(based on literature values; see main text). The white circles indicate the samples obviating the
expected correlation between soil $C_{org}$ and mud contents.

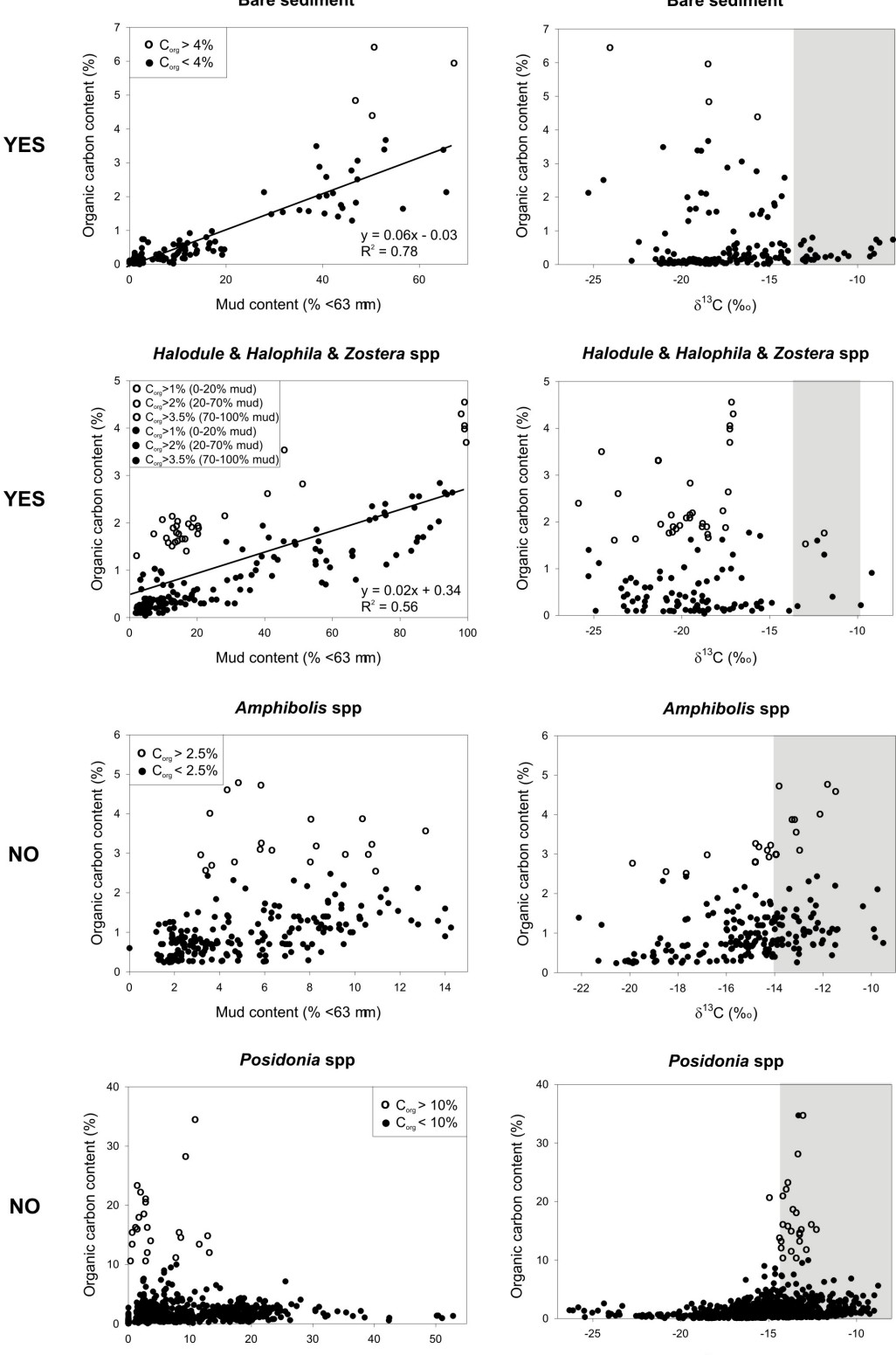

**Figure 2.** Relationships among soil $C_{org}$ and mud contents in 1 to 10 cm and 11 to 110 cm thick

580        soils: bare sediments, combined *Halodule*, *Halophila* and *Zostera* species, and combined

581        *Amphibolis* and *Posidonia* species. Only correlations with $R^2$ >0.5 are shown. The white

582        circles indicate the samples obviating the expected correlation between soil $C_{org}$ and mud

583        contents.

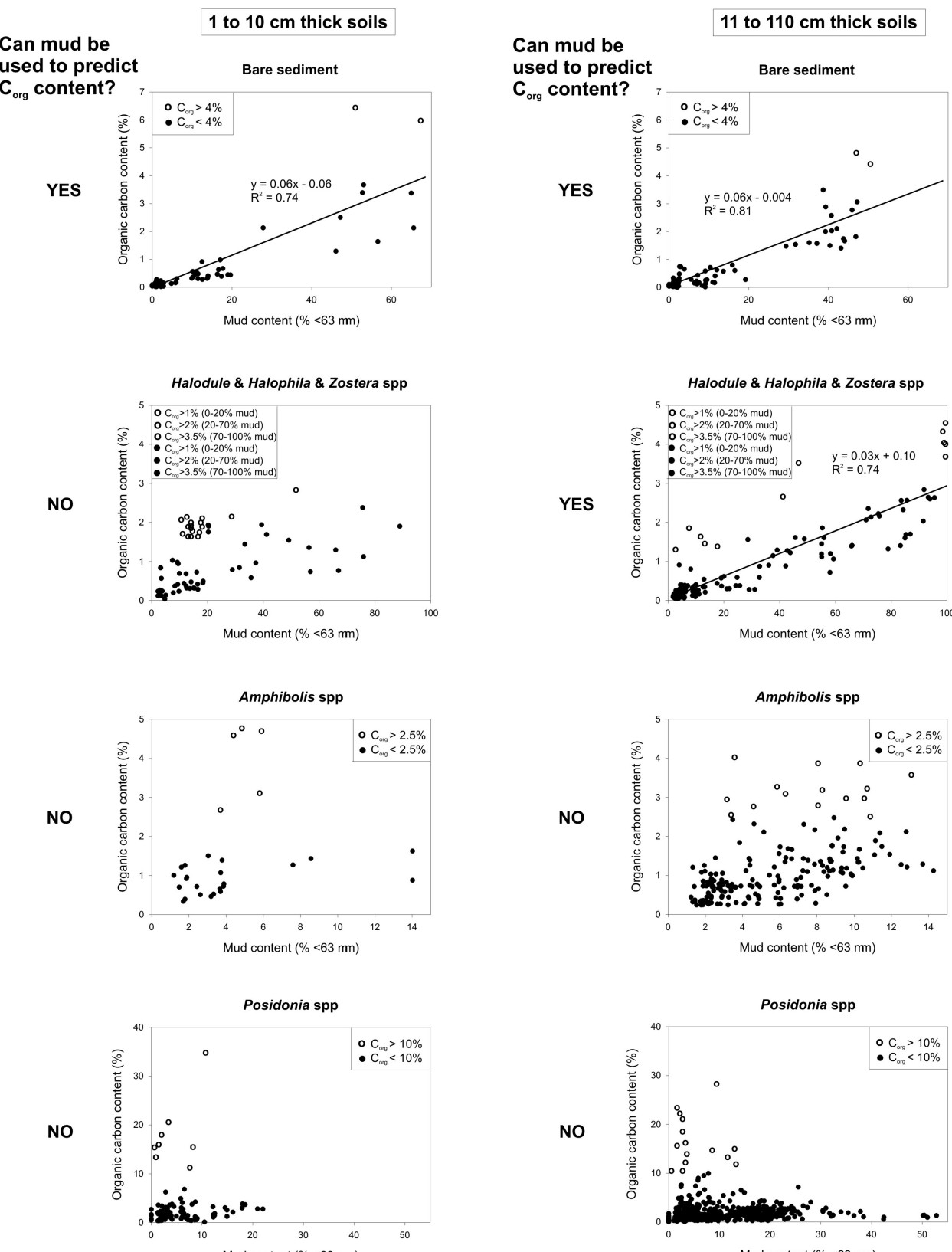



**Figure 3.** Relationships among soil C_org and mud contents, and soil C_org and $\delta^{13}$C signatures in
the coastal and estuarine habitats studied. The grey shaded areas showed the range of $\delta^{13}$C
signatures of plant detritus (based on literature values; see main text). The white circles
indicate the samples obviating the expected correlation between soil C_org and mud contents.

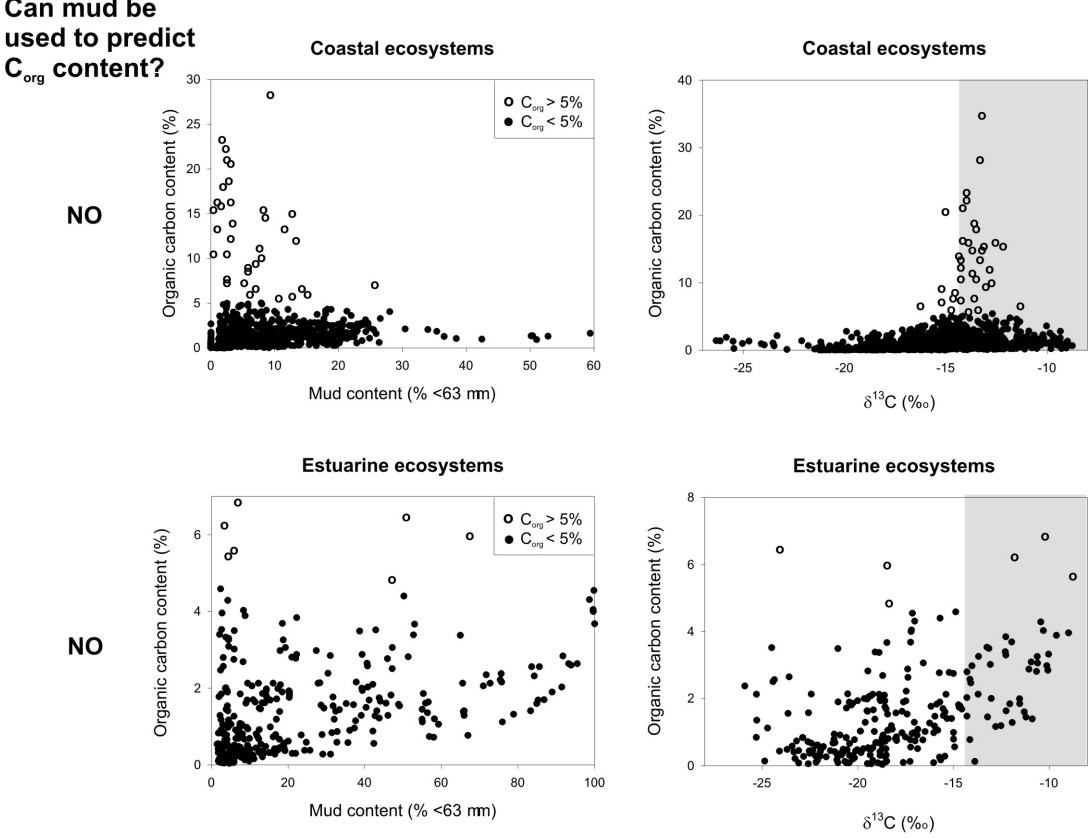


