# Peer review of "Can mud (silt and clay) concentration be used to predict soil organic carbon"

_Biogeosciences, 2015_

## Referee Comment (RC1) · Anonymous Referee #1 · 28 Jan 2016

Serrano et al review General Comments: This manuscript reports the organic matter content of seagrass sediments as a function of mud content and type of seagrass, focusing on whether percent mud offers predictive value for organic matter content. They find that sediments hosting longer-lived species, with greater below-ground biomass, accumulate organic matter to levels beyond that predicted by, and not well correlated to, mud content. The paper will thus be a useful report for "blue carbon" strategists in relating carbon accumulation to types of seagrass meadows, and fits the mission of BGD. To be useful, however, some clarifications are needed.

Specific Comments: 1. The relationship between organic matter and minerals is usually best seen with clay-sized fractions, rather than at the $63\mu$m cutoff used here.
[Figure]

Phrased another way, a correlation coefficient between organic matter and weight fraction of fine-grained sediment would be expected to improve as the latter parameter uses finer sized cutoffs. The main point of this paper derives from the significance of correlations, and use of finer-grained cutoffs might have led to higher significance levels than those which led to the manuscript's conclusions.

2. Cores were collected to depths ranging from 10-475 cm. Because below-ground biomass likely does not extend as deeply as carbon storage, it would be useful to report any depth relationships found. Because the authors allude frequently to the "blue carbon" justifications for their study, they should alert readers to these depth implications for two issues.

First, upper soil horizons of terrestrial soils – where discrete organic detritus persists - commonly show organic matter concentrations above those levels explained by finer-grained mineral association (e.g., Gami et al. 2009, Geoderma, 153:304, and Mayer and Xing 2001, SSSA 65:250). Thus, does enhanced carbon storage of certain seagrasses species extend to depths below the zone of living or recently dead biomass?

Second, if tests of the predictive value of mud content were made at 0-10 cm only or 100-110 cm only, would the conclusion in lines 181, 189-190 and elsewhere still apply? While the authors may not have sufficient data for as thorough an analysis as was done for entire cores, the manuscript would benefit greatly from any insights based on subsets of the data.

3. The blue carbon rationale of this manuscript also calls for some perspective on how much of the carbon sequestered is due to seagrass. Likely the organic matter associated with minerals would be buried wherever the minerals accumulate in the absence of seagrass meadows. It is the organic matter that is not associated with the fine-grained minerals – roughly the residual carbon levels in meadows above the regression line of the bare sediment – that represents the amount that is additionally sequestered due to seagrass. Estimates of organic carbon partitioned into these two

pools is made possible by the mud content parameter.

4. The authors give some attention to the roles of below-ground biomass and the lability of detritus of different size, but these points could use expansion. For example, papers such as Harrison and Mann (1975, L&O 20:924) showed loss rates dependence on detritus size. Also, Ember et al (1987, MEPS 36:33, especially their Figure 2) found that short form Spartina sediments showed elevated OC and del-13C values relative to tall form Spartina sediments; that finding does not seem consistent with the authors' argument that larger seagrasses lead to more seagrass carbon accumulation.

5. The figures would benefit from letters to indicate individual plots (A, B, etc.).

In the figure captions the authors use the term "intermittent" circles. On the manuscript I received there is no visual difference between the two types of red circles on individual plots, so other than by deduction it's not clear what is meant by "intermittent". Further, many of these "circles" are actually ellipses.

The terms "mud Corg saturation" and "low/high seagrass input" are confusing. Saturation of mud most likely explains the linear trend as seen in the Bare Sediment organic carbon vs. mud content regression line; the points above this line represent samples that are above this saturation level, but the authors have labeled them as "mud-Corg saturation".

The "circled" "mud-Corg saturation" points for the Halodule plots do not seem to include exactly the same subset of samples for the left and right plots. The "circled" "High seagrass input" data points in the two Amphibolis plots also are clearly not all the same samples, so why do they have the same label?

6. Line 160. Insert "variance in" between "the" and "trends".

7. Line 169. What is a "poor but slightly significant correlation"?

8. Line 195. A better way of phrasing this idea would be something like "providing more surface area and hence binding sites for Corg per weight of mineral". Also, I

don't understand intent of the phrase "increasing the available. . .for Corg aggregates" in line 196.

9. The sentence in lines 201-204 is confusing – the points above this regression are not well-explained by the regression.

10. Line 202. Are the authors claiming that the three data points with del-13C of -25 (Figure 1, upper right plot) are the same as the data points with highest OC in the Figure 1, upper left plot? That could be true for only the sample with 6.6% OC, but the other two points must be close to the regression line. Thus these latter two terrestrially influenced samples are close to saturation – i.e. predicted by grain size.

11. Line 210. "obviating" would be a better word than "ending".

In summary, the manuscript is concerned primarily with the predictability of organic matter content by mud content, and seeks further insight into controls on organic matter by separating out some variables. Its main point of better predictive value in bare sediment and short-lived seagrass meadow sediment, but not in longer-lived seagrass meadows, is reasonably well made. The manuscript becomes confusing, however, in the explanations of different organic matter sources and amounts making up the total organic loading.

---

## Referee Comment (RC2) · M. Plus (Referee) · 17 Feb 2016

Review report on the manuscript submitted to BiogeoSciences

Can mud (silt and clay) concentration be used to predict soil organic carbon content within seagrass ecosystems? By Serrano et al.

General comments The manuscript reports on the correlations between organic carbon (Corg), mud content (silt and clay fraction < 63 $\mu$m) and delta-13C in sediment cores sampled in a variety of temperate and tropical seagrass habitats. More broadly, the authors investigate if mud content can be used as a good proxy for Corg sediment contents of seagrass ecosystems. The manuscript is totally consistent with the scope

of this journal. The topic seems to me important, in the context of a growing interest in carbon sequestration assessment for marine ecosystems. The techniques used are not new, but this paper is well documented and refers, to my opinion, to relevant bibliography. The manuscript's main point (the mud content is not a universal proxy for blue carbon content but can be used for bare sediments and opportunistic seagrass ecosystems) is well supported by the observations as well as by the statistical treatment. I nevertheless have pointed out a few questions and comments that seem to me worthy to be answered before publication :

Specific comments / technical corrections

1. line 73 & 76 : I would prefer the words "significant relationship" instead "positive relationship". Even if it is true that we logically expect a positive relationship between mud content and Corg, rigorously a strong significant negative relationship could be as useful as a positive one.

2. Line 132-134 : This sentence is not true for P. oceanica. Table 3 shows that for that species, the Corg content decreases when the mud content increases.

3. In Table 2 : Amphibolis grifficiae or Amphibolis griffithii ?

4. Line 148 : the "exponential tendency" for combined Amphibolis spp. is speculative, please rephrase or test non linear relationships.

5. Lines 176 to 182 : This is confusing to me. You say before that ... ... fine-grained sediment can bind larger amount of Corg. But the capacity for silt and clay to bind Corg is limited. ... high mud content in sediments provide reducing conditions that can preserve Corg (lower mineralization rates). then why this could explain relative high Corg contents for some bare sediments with low mud contents ? This mud-Corg saturation needs to be clarified (specially for non-specialists as me).

6. Table 3 : please add in caption what na stands for (not available ?). Would ns – non significant – not be better ?

7. Figure 1 and 2 : I don't see any difference between the red and the red-intermittent circles in the manuscript version I received. Please, verify.

8. Figure 2, lower-left graph (Mud content vs Corg for estuarine ecosystems). There are 4 points showing high Corg contents (around 6%) for very low mud contents. To which type of ecosystem are they related ? P. autralis ?

---

## Author Comment (AC1) · 21 Apr 2016

Interactive comment on "Can mud (silt and clay) concentration be used to predict soil organic carbon content within seagrass ecosystems?" by O. Serrano et al.

O. Serrano et al. o.serranogras@ecu.edu.au

Response to Reviewers for "Can mud (silt and clay) concentration be used to predict soil organic carbon content within seagrass ecosystems?" by Serrano et al. We would like to thank the Reviewers for their efforts and comments, which have the potential to improve the manuscript. Please find below a detailed response to each of the issues raised.

Anonymous Referee #1 Serrano et al. review General Comments: This manuscript reports the organic matter content of seagrass sediments as a function of mud content and type of seagrass, focusing on whether percent mud offers predictive value for organic matter content. They find that sediments hosting longer-lived species, with greater below-ground biomass, accumulate organic matter to levels beyond that predicted by, and not well correlated to, mud content. The paper will thus be a useful report for "blue carbon" strategists in relating carbon accumulation to types of seagrass meadows, and fits the mission of BGD. To be useful, however, some clarifications are needed. Specific Comments: 1. The relationship between organic matter and minerals is usually best seen with clay-sized fractions, rather than at the $63\mu$m cutoff used here. Phrased another way, a correlation coefficient between organic matter and weight fraction of fine-grained sediment would be expected to improve as the latter parameter uses finer sized cutoffs. The main point of this paper derives from the significance of correlations, and use of finer-grained cutoffs might have led to higher significance levels than those which led to the manuscript's conclusions.

Response comment 1: In this manuscript we explored the relationships between mud (clay and silt, 'lutite'; <0.063 mm) and organic carbon (Corg) contents in bulk seagrass soils. There are three main issues or disadvantages linked to the use of clay (<0.004 mm) instead of clay and silt (mud, <0.063 mm) to explore the relationships between sediment grain-size composition and Corg content in bulk soils: A. Seagrass soils generally don't have much clay (concentrations ranging from 0 to 0.5%; in particular in meadows found in coastal areas). B. Data on clay concentration is not available for half of our analyses because our laser-diffraction particle analyser measured silt and clay (mud) together. C. The silt fraction (0.004-0.063 mm) provides binding sites for Corg on the surface of minerals, increasing the available space within the mineral matrix for Corg aggregates, and thereby potentially underestimating Corg content. Therefore, correlations between bulk soil Corg and mud sediment fraction (i.e. silt and clay) are expected to be higher than correlations between bulk soil Corg and clay (<0.004 mm) because: 1) the lower number of 'zeros' in the correlations; and 2) the mud fraction is

more representative of the bulk soil (i.e. silt also binds Corg) and a higher correlation is expected when comparing bulk soil Corg with a larger and more representative fraction of the sediment (clay and silt rather than clay alone). Indeed, the lack of clay data for a substantial portion of the dataset precludes a comprehensive exploration of the relationships.

2. Cores were collected to depths ranging from 10-475 cm. Because below-ground biomass likely does not extend as deeply as carbon storage, it would be useful to report any depth relationships found. Because the authors allude frequently to the "blue carbon" justifications for their study, they should alert readers to these depth implications for two issues. First, upper soil horizons of terrestrial soils – where discrete organic detritus persists - commonly show organic matter concentrations above those levels explained by finer- grained mineral association (e.g., Gami et al. 2009, Geoderma, 153:304, and Mayer and Xing 2001, SSSA 65:250). Thus, does enhanced carbon storage of certain sea- grasses species extend to depths below the zone of living or recently dead biomass? Second, if tests of the predictive value of mud content were made at 0-10 cm only or 100-110 cm only, would the conclusion in lines 181, 189-190 and elsewhere still apply? While the authors may not have sufficient data for as thorough an analysis as was done for entire cores, the manuscript would benefit greatly from any insights based on subsets of the data.

Response comment 2. We explored the relationships between soil Corg and mud contents within different core depths (from 1 to 10 cm-thick deposits, and from 11 to up to 475 cm thick deposits) for bare sediments and each group of seagrass species (Figure 1 below), as suggested by the referee. The results obtained show that soil depth is not an important factor when attempting to predict soil Corg content based on mud content in bare sediments (i.e. $R^2 > 0.74$ for all core depths explored; 1 to 110 cm-thick, 1 to 10 cm-thick, and 11 to 110 cm-thick). However, a clearer pattern appeared when exploring the correlation between soil Corg and mud contents in top 10 cm and within 11-110 cm soil depths of combined Halodule, Halophila and Zostera species ($R^2 = 0.17$ and

R2 = 0.74, respectively). These results suggest that the relatively small belowground biomass of these species (i.e. organic detritus) only has an impact on the expected positive correlation between soil Corg and mud content within the top 10 cm, while the correlation for deeper soil depths (11-110 cm) improved (R2 = 0.74) compared to the whole dataset (R2 = 0.56). For combined Amphibolis and Posidonia species, the results obtained show that soil depth is not an important factor when attempting to predict soil Corg content based on mud content (i.e. R2 <0.3 in all cases; whole dataset, 1 to 10 cm-thick, and 11 to 110 cm thick). These results suggest that the relatively large belowground biomass of these species (i.e. organic detritus) has an impact on the expected positive correlation between soil Corg and mud content within all depths (from 1 to 475 cm). All above could be included in the discussion of the final paper, after Editor's considerations.

3. The blue carbon rationale of this manuscript also calls for some perspective on how much of the carbon sequestered is due to seagrass. Likely the organic matter associated with minerals would be buried wherever the minerals accumulate in the absence of seagrass meadows. It is the organic matter that is not associated with the fine-grained minerals – roughly the residual carbon levels in meadows above the regression line of the bare sediment – that represents the amount that is additionally sequestered due to seagrass. Estimates of organic carbon partitioned into these two pools is made possible by the mud content parameter.

Response comment 3. Although it would be possible to run mixing models (i.e. based on the stable carbon isotopes rates of the organic matter explored in this study) to determine the percentage contribution of autochthonous (plant detritus) and allochthonous (seston, algae and terrestrial matter) Corg sources into the soil Corg pool for each ecosystem, we dismissed this option mainly because of the assumptions involved with this approach and its complexity (i.e. lack of robust $\delta$13C data of potential Corg sources for each site, thereby including $\delta$13C variability within latitude gradients, water depth, seasonality, diagenetic effects, etc.). The phenomena of Corg accumulation occurs over centennial time scales and we consider that using the same $\delta$13C values of potential sources for all sites to run the models could lead to misleading results. Instead, we preferred to show the range of $\delta$13C for seagrass detritus in Figure 1 of the manuscript, to allow identifying trends without running potentially misleading mixing models.

4. The authors give some attention to the roles of below-ground biomass and the lability of detritus of different size, but these points could use expansion. For example, papers such as Harrison and Mann (1975, L&O 20:924) showed loss rates dependence on detritus size. Also, Ember et al (1987, MEPS 36:33, especially their Figure 2) found that short form Spartina sediments showed elevated OC and del-13C values relative to tall form Spartina sediments; that finding does not seem consistent with the authors' argument that larger seagrasses lead to more seagrass carbon accumulation.

Response comment 4. In our paper we discussed differences in Corg storage and lability between seagrass groups. We argued that the contribution of seagrass detritus to the long-term carbon pool is a function of both sediment conditions for preservation and the intrinsic recalcitrance of the plant material itself, and plant productivity. Minor adjustments to the discussion will be made to include comparisons with other ecosystems (i.e. salt marshes), after Editor's considerations.

5. The figures would benefit from letters to indicate individual plots (A, B, etc.). In the figure captions the authors use the term "intermittent" circles. On the manuscript I received there is no visual difference between the two types of red circles on individual plots, so other than by deduction it's not clear what is meant by "intermittent". Further, many of these "circles" are actually ellipses. The terms "mud Corg saturation" and "low/high seagrass input" are confusing. Saturation of mud most likely explains the linear trend as seen in the Bare Sediment organic carbon vs. mud content regression line; the points above this line represent samples that are above this saturation level, but the authors have labeled them as "mud-Corg saturation". The "circled" "mud-Corg saturation" points for the Halodule plots do not seem to include exactly the same subset

of samples for the left and right plots. The "circled" "High seagrass input" data points in the two Amphibolis plots also are clearly not all the same samples, so why do they have the same label? 6. Line 160. Insert "variance in" between "the" and "trends". 7. Line 169. What is a "poor but slightly significant correlation"? 8. Line 195. A better way of phrasing this idea would be something like "providing more surface area and hence binding sites for Corg per weight of mineral". Also, I don't understand intent of the phrase "increasing the available. . .for Corg aggregates" in line 196. 9. The sentence in lines 201-204 is confusing – the points above this regression are not well-explained by the regression. 10. Line 202. Are the authors claiming that the three data points with del-13C of -25 (Figure 1, upper right plot) are the same as the data points with highest OC in the Figure 1, upper left plot? That could be true for only the sample with 6.6% OC, but the other two points must be close to the regression line. Thus these latter two terrestrially influenced samples are close to saturation – i.e. predicted by grain size. 11. Line 210. "obviating" would be a better word than "ending".

Response comments 5-11. We agree with the comments made by the referee and we are willing to address these corrections in the final version of the manuscript, after Editor's considerations.

In summary, the manuscript is concerned primarily with the predictability of organic matter content by mud content, and seeks further insight into controls on organic matter by separating out some variables. Its main point of better predictive value in bare sediment and short-lived seagrass meadow sediment, but not in longer-lived seagrass meadows, is reasonably well made. The manuscript becomes confusing, however, in the explanations of different organic matter sources and amounts making up the total organic loading.

Response: We are willing to address all your concerns in the final version of the manuscript, after Editor's considerations.

M. Plus (Referee #1) Review report on the manuscript submitted to BiogeoSciences

Can mud (silt and clay) concentration be used to predict soil organic carbon content within seagrass ecosystems? By Serrano et al. General comments The manuscript reports on the correlations between organic carbon (Corg), mud content (silt and clay fraction < 63 $\mu$m) and delta-13C in sediment cores sampled in a variety of temperate and tropical seagrass habitats. More broadly, the authors investigate if mud content can be used as a good proxy for Corg sediment contents of seagrass ecosystems. The manuscript is totally consistent with the scope of this journal. The topic seems to me important, in the context of a growing interest in carbon sequestration assessment for marine ecosystems. The techniques used are not new, but this paper is well documented and refers, to my opinion, to relevant bibliography. The manuscript's main point (the mud content is not a universal proxy for blue carbon content but can be used for bare sediments and opportunistic seagrass ecosystems) is well supported by the observations as well as by the statistical treatment. I nevertheless have pointed out a few questions and comments that seem to me worthy to be answered before publication: Specific comments / technical corrections 1. line 73 & 76: I would prefer the words "significant relationship" instead "positive relationship". Even if it is true that we logically expect a positive relationship between mud content and Corg, rigorously a strong significant negative relationship could be as useful as a positive one. 2. Line 132-134: This sentence is not true for P. oceanica. Table 3 shows that for that species, the Corg content decreases when the mud content increases. 3. In Table 2: Amphibolis grifficiae or Amphibolis griffithii? 4. Line 148: the "exponential tendency" for combined Amphibolis spp. is speculative, please rephrase or test non linear relationships. 5. Lines 176 to 182: This is confusing to me. You say before that fine-grained sediment can bind larger amount of Corg. But the capacity for silt and clay to bind Corg is limited, high mud content in sediments provide reducing conditions that can preserve Corg (lower mineralization rates). Then why this could explain relative high Corg contents for some bare sediments with low mud contents ? This mud-Corg saturation needs to be clarified (specially for non-specialists as me). 6. Table 3: please add in caption what na stands for (not available ?). Would ns – non significant – not be better? 7. Figure 1

and 2: I don't see any difference between the red and the red-intermittent circles in the manuscript version I received. Please, verify. 8. Figure 2, lower-left graph (Mud content vs Corg for estuarine ecosystems). There are 4 points showing high Corg contents (around 6%) for very low mud contents. To which type of ecosystem are they related? P. autralis?

Response comments. We agree with the specific comments and technical corrections made by the referee and we are willing to address these corrections and clarifications in the final version of the manuscript, after Editor's considerations.

Figure 1. Relationships among soil Corg and mud contents in the habitats studied: bare sediments, combined Halodule, Halophila and Zostera species, and combined Amphibolis and Posidonia species.
* * *
[Figure]

**Fig. 1.**

---

## Author Response (AR1)

*Response to the letter from the Editor on* **"Can mud (silt and clay) concentration be used to**
**predict soil organic carbon content within seagrass ecosystems?" by O. Serrano et al.**

**O. Serrano et al.**

**o.serranogras@ecu.edu.au**

Letter from the Editor:

Dear authors

Having now read your answers to the reviewer's comments and projected changes to the manuscript, I am happy to encourage you to proceed with the full revision of your manuscript. In addition to all the minor adjustments you have mentioned, I recommend you carefully address in your revised MS the following points raised by referee #1:

1. Briefly discuss the potential impact of the selected grain size cut off on your conclusions

2. Discuss the difference in correlations in surface and deep soils, including the additional figure

3. Improve the quality of the figures, if possible avoiding the "ellipses" or "circles", using rather different symbols and objective criteria for samples with "high/low seagrass inputs", etc…

Looking forward to reading this soon.

Best regards, Gwenaël Abril

Response:

Dear Dr Gwenaël Abril,

We would like to thank you for reviewing and handling our manuscript. We carefully addressed the three points raised by referee #1 in the new version submitted. All other minor comments raised by the two referees have been considered and included in the revised manuscript as suggested, unless stated otherwise. Please find below a detailed response to the comments raised
during the review process:

**Main adjustments**

• The potential impact of the selected grain size cut-off used in this study has been
discussed toward the end of the manuscript, as suggested.

[revised manuscript text omitted]

- We improved the quality of the figures by avoiding the "ellipses", and by using different symbols and objective criteria to identify samples with "high/low seagrass inputs" and "mud-Corg saturation", as suggested (Figures 1 to 3).

- We expanded the discussion on how detritus size could influence decay rates of seagrass detritus. We did not compare saltmarsh and seagrass ecosystems in terms of 1) decay rates dependence on detritus size and 2) relationships between plant size and OC content and d13C signatures. Instead, we used existing literature on seagrass to develop these topics in the discussion. We did not address in more detail how plant size (i.e. seagrass species) could influence OC constant and d13C values based because is not the focus of our article. However, Table 2 provides a comprehensive overview of the differences in OC storage and d13C signatures among seagrass species.

Text added (L308-L315): 'In addition, the larger size of detritus within *Amphibolis* and *Posidonia* meadows compared to *Halophila*, *Halodule* and *Zostera* meadows could also contribute to the larger accumulation of $C_{org}$ in the former, since decay rates of seagrass detritus increase with decreasing particle size due to larger surfaces available for microbial attack (Harrison, 1989)'

*Reference cited*: Harrison, P. G.: Detrital processing in seagrass systems: A review of factors affecting decay rates, remineralization and detritivory. Aquat. Bot., 263-288,

1989.

**Minor adjustments**

• *Line 160. Insert "variance in" between "the" and "trends".* Corrected as suggested.

• *Line 169. What is a "poor but slightly significant correlation"?* Clarified as suggested.

Text now reads (L197): 'a poor correlation in estuarine ecosystems'.

• *Line 195. A better way of phrasing this idea would be something like "providing more*

*surface area and hence binding sites for $C_{org}$ per weight of mineral". Also, I don't*

*understand intent of the phrase "increasing the available. . .for $C_{org}$ aggregates" in line*

*196.* Clarified as suggested.

Text now reads (L224-L228): 'The positive relationship found between mud and $C_{org}$

contents in coastal bare sediments (explaining 78% of the variability) is in agreement with previous studies (e.g. Bergamaschi et al. 1997; De Falco et al. 2004), and is related to their larger surface areas compared to coarse-grained sediments, providing more binding sites for $C_{org}$ on the surface of minerals'.

• *The sentence in lines 201-204 is confusing – the points above this regression are not*

*well-explained by the regression.* Clarified as suggested.

Text now reads (L238-L241): 'The results obtained showed that bare sediment samples with relative high $C_{org}$ contents (i.e. >4% $C_{org}$) and relatively low mud contents were also

$^{13}$C-depleted (Figure 1), suggesting significant contributions of soil $C_{org}$ from allochthonous sources (e.g. terrestrial and sestonic; Kennedy et al. 2010)'.

• *Line 202. Are the authors claiming that the three data points with del-13C of -25 (Figure*

*1, upper right plot) are the same as the data points with highest OC in the Figure 1,*

*upper left plot? That could be true for only the sample with 6.6% OC, but the other two*

*points must be close to the regression line. Thus these latter two terrestrially influenced*

*samples are close to saturation – i.e. predicted by grain size.* Clarified as suggested (see comment above).

• *Line 210. "obviating" would be a better word than "ending".* Corrected as suggested.

• *Line 73 & 76: I would prefer the words "significant relationship" instead "positive*

*relationship". Even if it is true that we logically expect a positive relationship between*

*mud content and $C_{org}$, rigorously a strong significant negative relationship could be as*

*useful as a positive one.* Corrected as suggested.

• *Line 132-134: This sentence is not true for P. oceanica. Table 3 shows that for that*

*species, the $C_{org}$ content decreases when the mud content increases.* Corrected as suggested.

Text now reads (L169-L172): 'the $C_{org}$ content increased with increasing mud content in bare sediments ($R^2 = 0.78$) and at species level, except for *Posidonia oceanica* (i.e. $C_{org}$

content decreased with increasing mud content; $R^2 = 0.15$) and *Amphibolis griffithii* (i.e.

no relationship was found, $R^2 = 0.05$; Table 3)'.

• *In Table 2: Amphibolis grifficiae or Amphibolis griffithii?* Clarified as suggested: the species is *Amphibolis griffithii*

• *Line 148: the "exponential tendency" for combined Amphibolis spp. is speculative,*

*please rephrase or test non linear relationships.* Corrected as suggested.

Text now reads (L201-L202): 'with a tendency of $C_{org}$-rich soils being enriched in $^{13}$C

(Figure 1)'.

• *Lines 176 to 182: This is confusing to me. You say before that fine-grained sediment can*

*bind larger amount of $C_{org}$. But the capacity for silt and clay to bind $C_{org}$ is limited, high*

*mud content in sediments provide reducing conditions that can preserve $C_{org}$ (lower*

*mineralization rates). Then why this could explain relative high $C_{org}$ contents for some*

*bare sediments with low mud contents ? This mud-$C_{org}$ saturation needs to be clarified*

*(specially for non-specialists as me).* Clarified for bare sediments, *Posidonia* and

*Amphibolis* meadows, and *Halodule, Halophila* and *Zostera* meadows.

• Text now reads:

L236-L241: 'However, the maximum capacity of a given soil to preserve $C_{org}$ by their association with clay and silt particles is limited (i.e. mud-$C_{org}$ saturation; Hassink, 1997).

The results obtained showed that bare sediment samples with relative high $C_{org}$ contents (i.e. >4% $C_{org}$) and relatively low mud contents were also $^{13}$C-depleted (Figure 1), suggesting significant contributions of soil $C_{org}$ from allochthonous sources (e.g.
terrestrial and sestonic; Kennedy et al. 2010)'.

L268-L274: 'The poor relationship between mud and soil $C_{org}$ contents in *Amphibolis*
soils could be explained by samples with relative high $C_{org}$ contents (i.e. >2.5% $C_{org}$) and
relatively low mud contents, as a result of both the contribution of seagrass-derived $C_{org}$
(i.e. $^{13}$C-enriched) and $C_{org}$ from allochthonous sources (i.e. $^{13}$C-depleted; Figure 1). In
*Posidonia* soils, the poor relationship between mud and soil $C_{org}$ contents could be
explained by samples with relative high $C_{org}$ contents (i.e. >10% $C_{org}$) and relatively low
mud contents, as a result of the contribution of seagrass-derived $C_{org}$ (i.e. $^{13}$C-enriched;
Figure 1)'.

L280-L287: 'The positive relationship between mud and soil $C_{org}$ contents in *Halodule,*
*Halophila and Zostera* soils could be explained their relatively high mud content and $^{13}$C-
depleted $C_{org}$, indicating that allochthonous $C_{org}$ inputs and mud content play a major role
in soil $C_{org}$ accumulation in these opportunistic and early-colonizing seagrasses.
However, the relative high $C_{org}$ contents found with relatively low mud contents (i.e.
mud-$C_{org}$ saturation) disrupted the correlation found between soil $C_{org}$ and mud contents
in these meadows ($C_{org}$ >1% in samples with 0-20% mud; $C_{org}$ >2% in samples with 20-
70% mud and $C_{org}$ >3.5 in samples with 70-100% mud; Figure 1)'.

• *Table 3: please add in caption what na stands for (not available ?). Would ns – non*
*significant – not be better?* Corrected as suggested.

• *Figure 1 and 2: I don't see any difference between the red and the red-intermittent*
*circles in the manuscript version I received. Please, verify.* Clarified as suggested (see
above).

• *Figure 2, lower-left graph (Mud content vs $C_{org}$ for estuarine ecosystems). There are 4*
*points showing high $C_{org}$ contents (around 6%) for very low mud contents. To which type*
*of ecosystem are they related? P. autralis?* These four points belong to estuarine *P.*
*australis* meadows; three out of the four samples contained large amounts of seagrass-
derived $C_{org}$ (white circles in the top-right of the lower-right graph in Figure 3).

MANUSCRIPT WITH CHANGES TRACKED:

[revised manuscript text omitted]

Oscar Serrano 28/5/2016 1:52 PM

a)

| Habitat (species) | For |
|---|---|
| *Posidonia oceanica* | $C_{org}$ |
| *Posidonia australis* | $C_{org}$ |
| *Posidonia sinuosa* | $C_{org}$ |
| *Amphibolis* (mixed spp) | $C_{org}$ |
| *Amphibolis antarctica* | $C_{org}$ |
| *Amphibolis griffithii* | |
| *Halodule uninervis* | $C_{org}$ |
| *Zostera muelleri* | $C_{org}$ |
| *Halophila ovalis* | $C_{org}$ |
| Bare | $C_{org}$ |

b)

| Habitat (geomorphology) | For |
|---|---|
| Coastal | |
| Estuarine | $C_{org}$ |

**Figure 1.** Relationships among soil $C_{org}$ and mud contents, and soil $C_{org}$ and $\delta^{13}C$ signatures in all habitats and all soil depths studied: bare sediments, combined *Halodule*, *Halophila* and

*Zostera* species, and combined *Amphibolis* and *Posidonia* species. Only correlations with $R^2$

>0.5 are showed. The grey shaded areas showed the range of $\delta^{13}C$ signatures of plant detritus (based on literature values; see main text). The white circles indicate the samples obviating the expected correlation between soil $C_{org}$ and mud contents.

[Figure]

**Can mud be used to predict $C_{org}$ content?**

**Figure 2.** Relationships among soil $C_{org}$ and mud contents in 1 to 10 cm and 11 to 110 cm thick soils: bare sediments, combined *Halodule*, *Halophila* and *Zostera* species, and combined

*Amphibolis* and *Posidonia* species. Only correlations with $R^2$ >0.5 are showed. The white circles indicate the samples obviating the expected correlation between soil $C_{org}$ and mud contents.

[Figure]

**Figure 3.** Relationships among soil $C_{org}$ and mud contents, and soil $C_{org}$ and $\delta^{13}C$ signatures in the coastal and estuarine habitats studied. The grey shaded areas showed the range of $\delta^{13}C$ signatures of plant detritus (based on literature values; see main text). The white circles indicate the samples obviating the expected correlation between soil $C_{org}$ and mud contents.

[Figure]

---

## Author Response (AR2)

- Response to the letter from the Editor and Referee #3 on "Can mud (silt and clay) 1
- concentration be used to predict soil organic carbon content within seagrass ecosystems?" 2
- 3 by O. Serrano et al.
- 4
- 5 O. Serrano et al.

o.serranogras@ecu.edu.au 6

7

**8 Letter from the Editor:**

Dear Authors, 9

I have read the revised version of your revised MS. In the meanwhile, I asked for a third 10 reviewer to evaluate your revised MS, and her/his comments are attached. We both agree that 11 your MS should be published after considering several aspects. Please respond to the comment 12 by the new reviewer and modify your MS accordingly. In particular, the reviewer is asking why 13 it is relevant and interesting to measure grain size rather than OC content. This seems to me quiet 14

important, as all you MS, including the title, is constructed on that postulate, in a blue carbon 15

perspective. Alternatively, predicting OC content from grain size is probably not the only 16

relevant message of your paper. Please clarify this point. 17

In addition, I was a bit disappointed by the criteria you have chosen to separate high/low 18

seagrass samples in the figures: OC content is trivial as this parameter appears on the Y axis of 19 both panels. Please try with d13C.

20

I am looking forward reading a revised version of you MS as well as a detailed response to these 21

- comments 22
- All the best 23
- Gwen Abril 24
- 25
- **Response:** 26
- 27 Dear Dr Gwenaël Abril,

28 We would like to thank you for reviewing and handling our manuscript. We carefully addressed

the points raised by referee #3 and yourself in the new version submitted. Please find below a detailed response to the comments raised during the review process.

31

32 1. The analyses of soil grain size (i.e. %mud) could constitute a relatively cheap method to estimate soil organic carbon content in seagrass ecosystems, particularly dry and wet sieving 33 using standard geological sieves (Erftemeijer and Kach, 2001). These could be used to cheaply 34 35 quantify mud content as a proxy for carbon, particularly in student projects, citizen science and in countries where funding for science is limited and they do not have access to higher 36 technology methods or cannot afford to pay for analysis. Indeed, maps of soil grain size 37 distribution are available for several areas and regions (e.g. Passlow et al. 2005) or could be 38 obtained using remote sensing (Rainey et al. 2003; De Falco et al. 2010), opening new 39 opportunities for scaling exercises. This topic was partially addressed in the last paragraph of the 40 41 introduction:

"A significant relationship between mud and Corg contents would allow mud to be used as a 42 43 proxy for Corg content, thereby enabling robust scaling up exercises at a low cost as part of blue carbon stock assessments. Furthermore, since most countries have conducted geological surveys 44 within the coastal zone to determine sediment grain size, a strong, positive relationship between 45 mud and Corg contents would allow the development of geomorphology models to predict blue 46 carbon content within seagrass meadows, dramatically improving global estimates of blue carbon 47 storage. The purpose of this study was therefore to test for relationships between Corg and mud 48 contents within seagrass ecosystems and adjacent bare sediments." 49

and the discussion also referred to the main goals of our study:

51 "Overall mud content is a poor predictor of soil  $C_{org}$  in seagrass meadows and care should be 52 taken in its use as a cost-effective proxy or indicator of  $C_{org}$  for scaling-up purposes in the 53 emerging field of blue carbon science."

54 "...allow mud to be used as a proxy for Corg content in these ecosystems, thereby enabling robust 55 scaling up exercises (i.e. benefiting from existing geological surveys and models) at low cost as 56 part of blue carbon stock assessment programs."

57

58 In order to reinforce the significance and relevance of the findings in our study we included

59 further remarks along the manuscript:

60 Text added in the Abstract (L211-212): "The results obtained could enable robust scaling up

61 exercises at a low cost as part of blue carbon stock assessments."

Text added in the Discussion (L491-503): "Analyses of soil grain size (i.e. %mud) could 62 constitute a relatively cheap method to estimate soil organic carbon content in seagrass 63 ecosystems, particularly dry and wet sieving using standard geological sieves (Erftemeijer and 64 65 Kach, 2001). These could be used to cheaply quantify mud content as a proxy for carbon, particularly in student projects, citizen science and in countries where funding for science is 66 limited and they do not have access to higher technology methods or cannot afford to pay for 67 analysis. In addition, since most countries have conducted geological surveys within the coastal 68 zone to determine sediment grain size (e.g. Passlow et al. 2005), a strong, positive relationship 69 between mud and Corg contents could allow the development of geomorphology models to 70 71 predict blue carbon content within seagrass meadows, dramatically improving global estimates of blue carbon storage. Indeed, maps of soil grain-size could be obtained using remote sensing 72 (Rainey et al. 2003; De Falco et al. 2010), opening new opportunities for scaling exercises." 73

74

75 References:

- De Falco, G., Tonielli, R., Di Martino, G., Innangi, S., Simeone, S. and Parnum, I.M.:
  Relationships between multibeam backscatter, sediment grain size and Posidonia oceanica
  seagrass distribution. Continental Shelf research 30, 1941-1950, 2010.
- 79 Erftemeijer, P.L. and Koch, E.W.: Sediment geology methods for seagrass habitat. In: Short FT,
- 80 Coles RG (eds) Global seagrass research methods. pp 345-367, 2001.

81 Passlow, V., Rogis, J., Hancock, A., Hemer, M., Glenn, K. and Habib, A.: Final Report, National

82 Marine Sediments Database and Seafloor Characteristics Project. Geoscience Australia, Record

- 83 2005/08, 2005.
- Rainey, M.P., Tyler, A.N., Gilvear, D.J., Bryant, R.G. and McDonald, P.: Mapping intertidal
- 85 estuarine sediment grain-size distributions through airbone remote sensing. Remote Sensing of

- 86 Environment 86, 480-490, 2003.
- 87

2. The y-axis in Figures 1 to 3 correspond to the OC content, while x-axis correspond to %mud 88 content and therefore, one could use the formula (y=ax+b) to estimate OC content based on 89 %mud content. The d13C values were only used to determine whether soils with relatively high 90 91 OC content with respect to their mud content had relatively high seagrass inputs. This was illustrated by shading the range of d13C signatures of seagrass tissues in the Figures, to conclude 92 that allochthonous OC inputs play a major role in soil OC accumulation in opportunistic and 93 early-colonizing seagrasses (Halodule, Halophila and Zostera), but high-biomass and persistent 94 95 meadows (i.e. Posidonia and Amphibolis) accumulate higher seagrass-derived OC compared to ephemeral and low-biomass meadows. We do not consider the use of d13C appropriate to 96 highlight (i.e. empty circles) soils with high seagrass-derived OC in the Figures showing the 97 relationship between OC and mud content, rather highlight soils with high OC content to 98 evaluate their origin (i.e. high autochthonous or allochthonous OC). 99

100

**101 **Report from Referee #3:**

Overall, the premise of the paper is good, if mud content may be used as a proxy for Corg 102 sediment contents of seagrass ecosystems and adjacent bare sediments. However I have one 103 104 major concern: Instead of carbon content, have the authors compared the mud fraction carbon density (g cm-3) instead of carbon content? The reason I say this is that while carbon content 105 may decrease downcore, sediment density often increases downcore. As such, carbon density 106 107 may be fairly constant downcore (for instance see Donato et al. (2011)). Furthermore, sediment density is directly related to grain size and showing the carbon densities should reduce the effect 108 of soil depth and aging. The authors should have the dry bulk density since they have the dry 109 weights and volume for each interval. Therefore, I would suggest that the mud fraction would 110 have a greater relation to carbon density. Moreover, carbon density is a better link to blue carbon 111 112 (carbon accumulation) than carbon content.

Donato, D.C., Kauffman, J.B., Murdiyarso, D., Kurnianto, S., Stidham, M., Kanninen, M., 2011.
Mangroves among the most carbon-rich forests in the tropics. Nature Geoscience 4, 293-297.

115

116 **Response:**

117 Dear Referee,

- 118 We would like to thank you for reviewing our manuscript. We consider appropriate to keep our
- initial approach: compare %OC with %mud instead of g OC with %mud as suggested by the reviewer. %OC and %mud are easy to estimate, but estimate density and explore the relationships between g OC and %mud it is complex and out of scope in our study:
- 122 1. Changes in density with depth are not only related to %mud but also related to porosity and 123 sediment grain-size distribution. For example, sands have higher density but high porosity, so if 124 mud is present will fill empty spaces, diminishing porosity and increasing density. There exist 125 multiple combinations related to grain-size distribution that could affect density and were out of
- scope in our study.
- 2. Changes in density with depth are related to soil compaction linked to OC decomposition with
   ageing, and compression during coring and core processing (i.e. extrusion). Compression was not
   measured for several cores used in this study.
- 130

In summary, normalizing %OC by density could entail confusion and misleading conclusions, and it is not possible to explore the hypothesis suggested above because soil compression during coring and extrusion was not measured for several cores, and therefore it is not possible to 'decompress' the density values obtained before estimating g OC cm-3. Soil compression of loose soils (i.e seagrass meadows) during coring is an inevitable phenomenon and could entail up to 50% core shortening and large uncertainties/errors when exploring the relationships between g OC and %mud as suggested by the reviewer.

138

**139 **Comment from Referee #3:**

Another objective of this work is reduce costs in blue carbon research. Have the authors compared the cost in analyzing carbon content as compared to grainsize analyses? I would suggest that it is easier and cheaper measure of carbon content (and carbon density) than grain size analyses (sand, silt and clay fractions).

144

**145 **Response:**

The costs of soil OC content analysis range from \$12 to \$40, while sediment grain size analyses (i.e. %mud) by dry or wet sieving could be done at zero cost in any lab, being particularly useful in student projects, citizen science and in countries where funding for science is limited and they

- 149 do not have access to higher technology methods or cannot afford to pay for analysis. Indeed,
- 150 standard geological sieves are relatively cheap and commonly found in most laboratories.
- 151
- 152 Minor comment; the authors explain that mud content is composed of silt and clay too many time
- throughout the text. This only needs to be stated once in the abstract and once in the main text.

- 154 **Response:**
- 155 Redundancy was deleted as suggested
- 156
- 157
- 158

**159 Can mud (silt and clay) concentration be used to predict soil organic carbon**

**160 content within seagrass ecosystems?**

161 Oscar Serrano1,2\*, Paul S. Lavery1,3, Carlos M. Duarte4, Gary A. Kendrick2,5, Antoni Calafat6,

162 Paul York7, Andy Steven8, Peter Macreadie9,10

- 163

[revised manuscript text omitted]

- 727 Table 3. Pearson correlation analyses to test for significant relationships among soil Corg and
- mud contents, and soil  $C_{org}$  and  $\delta^{13}C$  signatures in up to 475 cm long cores; based on (a) species

identity and (b) habitat geomorphology. *ns*, non significant correlation.

| a)                     |                              |                |         |                                             |                  |         |
|------------------------|------------------------------|----------------|---------|---------------------------------------------|------------------|---------|
| Habitat                | Organic carbon (%) v         | 's mud         | (%)     | Organic carbon (%) v                        | $s \delta^{13}C$ | (‰)     |
| (species)              | Formula                      | $\mathbb{R}^2$ | P value | Formula                                     | $\mathbb{R}^2$   | P value |
| Posidonia oceanica     | $C_{org} = -0.26*mud + 6.95$ | 0.15           | ***     | $C_{org} = 1.59 * \delta^{13}C + 27.61$     | 0.13             | ***     |
| Posidonia australis    | $C_{org} = 0.02*mud + 1.69$  | 0.02           | *       | $C_{org} = 0.18*\delta^{13}C + 4.73$        | 0.30             | ***     |
| Posidonia sinuosa      | $C_{org} = 0.07*mud + 0.61$  | 0.09           | ***     | $C_{\rm org} = 0.12^* \delta^{13} C + 2.44$ | 0.23             | ***     |
| Amphibolis (mixed spp) | $C_{org} = 0.17*mud + 0.61$  | 0.26           | ***     | $C_{\rm org} = 0.14*\delta^{13}C + 3.53$    | 0.09             | **      |
| Amphibolis antarctica  | $C_{org} = 0.08*mud + 0.47$  | 0.32           | ***     | $C_{\rm org} = 0.14*\delta^{13}C + 3.10$    | 0.29             | ***     |
| Amphibolis griffithii  | ns                           | 0.05           | 0.18    | $C_{\rm org} = 0.06*\delta^{13}C + 1.79$    | 0.21             | **      |
| Halodule uninervis     | $C_{org} = 0.02*mud + 0.37$  | 0.34           | ***     | ns                                          | 0.00             | 0.89    |
| Zostera muelleri       | $C_{org} = 0.02*mud + 0.54$  | 0.39           | ***     | ns                                          | 0.08             | 0.07    |
| Halophila ovalis       | $C_{org} = 0.04*mud + 0.12$  | 0.91           | ***     | ns                                          | 0.00             | 0.89    |
| Bare                   | $C_{org} = 0.06*mud - 0.03$  | 0.78           | ***     | ns                                          | 0.01             | 0.24    |

| b)              |                             |                |         |                                             |                |         |
|-----------------|-----------------------------|----------------|---------|---------------------------------------------|----------------|---------|
| Habitat         | Organic carbon (%)          | vs mud         | (%)     | Organic carbon (%)                          | vs δ¹³C        | (‰)     |
| (geomorphology) | Formula                     | $\mathbb{R}^2$ | P value | Formula                                     | $\mathbb{R}^2$ | P value |
| Coastal         | ns                          | 0.01           | 0.85    | $C_{\rm org} = 0.17^* \delta^{13} C + 4.14$ | 0.03           | ***     |
| Estuarine       | $C_{org} = 0.02*mud + 1.01$ | 0.14           | *       | $C_{\rm org} = 0.17*\delta^{13}C + 4.52$    | 0.22           | **      |

Figure 1. Relationships among soil  $C_{org}$  and mud contents, and soil  $C_{org}$  and  $\delta^{13}C$  signatures in all habitats and all soil depths studied: bare sediments, combined *Halodule*, *Halophila* and *Zostera* species, and combined *Amphibolis* and *Posidonia* species. Only correlations with R2 >0.5 are shown. The grey shaded areas showed the range of  $\delta^{13}C$  signatures of plant detritus (based on literature values; see main text). The white circles indicate the samples obviating the expected correlation between soil  $C_{org}$  and mud contents.

Oscar Serrano 5/8/2016 2:30 PM Deleted: ed